# Intellectual Disability and Brain Creatine Deficit: Phenotyping of the Genetic Mouse Model for GAMT Deficiency

**DOI:** 10.3390/genes12081201

**Published:** 2021-08-02

**Authors:** Luigia Rossi, Francesca Nardecchia, Francesca Pierigè, Rossella Ventura, Claudia Carducci, Vincenzo Leuzzi, Mauro Magnani, Simona Cabib, Tiziana Pascucci

**Affiliations:** 1Department of Biomolecular Sciences, University of Urbino “Carlo Bo”, 61029 Urbino, Italy; luigia.rossi@uniurb.it (L.R.); francesca.pierige@uniurb.it (F.P.); mauro.magnani@uniurb.it (M.M.); 2EryDel SpA, Via Sasso 36, 61029 Urbino, Italy; 3Division of Child Neurology and Psychiatry, Department of Human Neuroscience, Sapienza University, 00185 Rome, Italy; francesca.nardecchia@uniroma1.it (F.N.); vincenzo.leuzzi@uniroma1.it (V.L.); 4Department of Psychology and “Daniel Bovet” Center, Sapienza University, 00184 Rome, Italy; rossella.ventura@uniroma1.it (R.V.); simona.cabib@uniroma1.it (S.C.); 5IRCCS Fondazione Santa Lucia, 00142 Rome, Italy; 6Department of Experimental Medicine, Sapienza University, 00161 Rome, Italy; claudia.carducci@uniroma1.it

**Keywords:** GAMT deficiency, genetic mouse model, behavioral phenotyping, developmental delay

## Abstract

Guanidinoacetate methyltransferase deficiency (GAMT-D) is one of three cerebral creatine (Cr) deficiency syndromes due to pathogenic variants in the GAMT gene (19p13.3). GAMT-D is characterized by the accumulation of guanidinoacetic acid (GAA) and the depletion of Cr, which result in severe global developmental delay (and intellectual disability), movement disorder, and epilepsy. The GAMT knockout (KO) mouse model presents biochemical alterations in bodily fluids, the brain, and muscles, including increased GAA and decreased Cr and creatinine (Crn) levels, which are similar to those observed in humans. At the behavioral level, only limited and mild alterations have been reported, with a large part of analyzed behaviors being unaffected in GAMT KO as compared with wild-type mice. At the cerebral level, decreased Cr and Crn and increased GAA and other guanidine compound levels have been observed. Nevertheless, the effects of Cr deficiency and GAA accumulation on many neurochemical, morphological, and molecular processes have not yet been explored. In this review, we summarize data regarding behavioral and cerebral GAMT KO phenotypes, and focus on uncharted behavioral alterations that are comparable with the clinical symptoms reported in GAMT-D patients, including intellectual disability, poor speech, and autistic-like behaviors, as well as unexplored Cr-induced cerebral alterations.

## 1. Introduction

In the brain, the creatine/phosphocreatine/creatine kinase system is essential in maintaining the high-energy phosphate levels necessary for central nervous system (CNS) development and functioning by regenerating, transporting, and buffering (adenosine triphosphate) ATP levels. Apart from this fundamental role in energy metabolism, creatine has been proposed as a neurotransmitter, since it is released from neurons in an action-potential-dependent manner, and acts as an agonist of postsynaptic γ aminobutyric acid (GABA)-A receptors. Moreover, creatine has been reported to be one of the main cellular osmolytes in the CNS, and may be a potential regulator of appetite and weight in the hypothalamic nuclei. These and other functions of creatine in the CNS are well summarized and described [1]. As previously reported, the identification of creatine deficiency syndromes (CDS) caused by mutations in the L-arginine:glycine amidinotransferase (*AGAT*) [2], guanidinoacetate methyltransferase deficiency (GAMT) [3], and *SLC6A8* genes [4] highlights the essential role of intact creatine metabolism in psychomotor development and cognitive function in humans.

Animal models of *AGAT*, GAMT, and *SLC6A8* deficiency have been successfully developed, and provide a unique opportunity to study specific aspects of these rare pathologies in depth and develop targeted therapeutic strategies. Since GAMT deficiency (GAMT-D) is considered to be the most severe CDS, we summarize data regarding biochemical and behavioral phenotypes of GAMT knockout (KO) mice in order to highlight similarities and possible differences between the mouse model and human pathology, and the behavioral alterations comparable with clinical symptoms reported in GAMT-D patients.

Though GAMT-D animals exhibit biochemical phenotyping (tissue and body fluid metabolite profiling) that parallels the biochemical phenotype of human patients, these animals develop only a mild cognitive deficit, in contrast to the severity seen in humans [5]. After an uneventful pregnancy and delivery, GAMT-D patients show normal early postnatal development followed by developmental arrest that progresses to severe intellectual disability without speech, intractable epileptic seizures, and movement disorder [6].

## 2. GAMT-D Syndrome: Clinical and Biochemical Profiling

Cerebral CDS (CCDS), inborn errors of creatine metabolism, include the two creatine biosynthesis disorders—*AGAT* deficiency, and GAMT-D—and the creatine transporter (CRTR) deficiency. The two creatine synthesis defects (*AGAT* and GAMT deficiencies) have an autosomal recessive inheritance pattern, while CRTR deficiency is inherited and X-linked [7,8,9].

GAMT-D (EC 2.1.1.2) was first described in 1994 in a 22-month-old boy with cerebral creatine deficiency, developmental delay, and progressive extrapyramidal symptoms [3,10]. Low urinary creatinine excretion, reflecting a low body creatine pool, and accumulation of guanidinoacetate—the substrate of the deficient enzyme activity—were characteristic biochemical findings [11]. Approximately 130 patients with GAMT-D have been reported in literature, either as individual case reports or as small case series with an incidence of 1:550,000–1:2,640,000 [12,13,14,15].

The biochemical signature of GAMT-D is the increase in guanidinoacetic acid (GAA) concentration associated with low levels of creatine in bodily fluids (urine, plasma, and cerebrospinal fluid (CSF)) and the absence of (or very low) a creatine peak on proton magnetic resonance spectroscopy (H1-MRS) [6,7]. Urine GAA levels are about 1–30-fold, plasma GAA levels are 2–20-fold (10-fold on average), and CSF GAA levels are 5–50-fold higher than the upper limit of the GAA reference range [16,17,18].

The diagnosis is confirmed after identification of biallelic pathogenic variants in *GAMT*. More than 70 pathogenic variants have been described in GAMT-D patients, and about half of them are missense variants. Deletions, splice errors, frame shift, nonsense, and truncating mutations have also been reported [6,7,17,19]. Interestingly, in Portuguese patients, a high prevalence of the c.59G>C pathogenic variant suggests a founder effect in this region [20].

Intellectual disability—the clinical hallmark of GAMT deficiency—is present in all affected patients, and is moderate or severe in most patients (60–90%) [7,17,21]. Less frequently, developmental arrest and regression is reported [12,17,22].

The second consistent manifestation of the disease is epilepsy, which may be drug-resistant in a small proportion of patients [6,17,19,23]. Febrile seizures may occur in the early phase of the disease, especially between 3 and 6 months of life, with epilepsy onset rarely delayed. There is no specific seizure pattern in GAMT-D, and the same patient can have different types of seizures at different ages. In the first months of life, life-threatening tonic seizures with apnea or myoclonic seizures have been reported, while in infancy and childhood myoclonic astatic seizures, generalized tonic–clonic seizures, partial seizures with secondary generalization, drop attacks, atypical absences, and staring episodes are more frequent. In GAMT-D, no typical electroencephalography (EEG) pattern has been defined. An early derangement of background organization, interictal multifocal spikes, and slow wave discharges have been described, as well as focal abnormalities with prominent frontal region involvement [17,23].

The third most frequent manifestation of GAMT-D is movement disorder, occurring in about half of patients—usually the most severe patients [17,21]. Movement disorder commonly consists of an extrapyramidal syndrome, mainly a hyperkinetic movement disorder. Dystonia and ataxia are the most common types, followed by chorea, choreoathetosis, ballismus, tremor, and myoclonus. Rett-like stereotypic movements of the hands have also been reported [24,25]. Bradykinesia and spasticity have been further associated with the disease [6,17,19,24,26]. Fever-induced ataxia lasting for days and slowly remitting after the illness has been described in only one patient [6]. Movement disorder is usually an early feature of GAMT-D, usually emerging before 12 years of age [21], although a late onset has been reported in a few patients [27,28]. In most—but not all—patients with movement disorder, bilateral pallidal lesions on brain magnetic resonance imaging (MRI) have been observed (hypointensity in T1-weighted and hyperintensity in T2-weighted images) [6,19,21,23]. In some patients, white matter alterations in the brainstem and pontine region, enlarged ventricles, and cerebral atrophy, in addition to basal ganglia lesions, have been reported [17,24,29].

In GAMT-D, a progressive course has been suggested, especially with regard to motor features. In some late-diagnosed patients, progressive dystonia and action tremor, spasticity, and dystonic–ballistic movements emerging during the second decade of life have been reported, raising the possibility of a neurodegenerative nature of the disease [17,21]. However, the natural history of the disease is not known, due to its rarity and the initiation of specific treatment that improves clinical features.

Language seems to be especially affected in GAMT-D. Most patients never develop speech, or may speak fewer than 10 single words [13,21,27,30,31]. Only one verbal patient has been reported in the literature—a 13-year-old girl who was able to speak in short sentences and use syntax, pronouns, negatives, and tenses [27].

Behavioral disorder is frequently associated with GAMT deficiency, mainly consisting of autistic features, hyperactivity, and aggressive and self-injurious behavior [19,21]. Many patients are described as lacking eye contact, having difficulties with social communication or socio-adaptive skills, not pointing fingers at objects, and rarely imitating actions [14,15,22]. Only three patients have been described as having a happy predisposition and frequent bursts of laughter [13].

A muscle phenotype of GAMT-D patients is not defined, but muscle mass and strength do not seem to be affected, other than hypotonia [17,19,26]. Asthenia was reported in one patient [32], while another patient was described as slim but strong [25]. The treatment of GAMT-D, aimed at restoring brain creatine (Cr) (by Cr supplementation) and at preventing GAA accumulation by substrate restriction (by ornithine and sodium benzoate supplementation and arginine restriction) [33], has been proven to be effective in presymptomatic patients [33,34,35], although some doubts have been raised over the usefulness of sodium benzoate supplementation and an arginine-restricted diet [35,36]. However, the late start of the treatment improves pharmacological control of seizures, but has limited effect in ameliorating the neurological deficits [6,17,21].

## 3. GAMT KO Mouse Model

Since systematic patient studies on the pathophysiology of CDS cannot be conducted due to the low incidence of this pathology, genetically-engineered mice represent an excellent experimental tool to investigate this disorder.

The generation of a KO mouse model for GAMT-D was reported for the first time in 2000 [37], and validated to study creatine deficiency in 2003 [38]. In a very elegant study [39], the authors showed that human and mouse tissues have comparable GAMT mRNA and protein expression patterns, clearly supporting the usefulness of this mouse model to study GAMT-D. They generated and described in detail a GAMT KO mouse model via targeted disruption of the open reading frame of the murine *GAMT* gene in the first exon in embryonic stem cells, which resulted in the creation of a null allele, as verified at the genetic, RNA, and protein levels. In the same work, the authors reported that GAMT KO mice show biochemical features in the serum, urine, brain, skeletal muscle, and heart that mirror many of the biochemical hallmarks of human disease. Indeed, they observed a marked increase in GAA and reduced Cr and creatinine (Crn) levels in the serum, urine, and brains of *GAMT*-/- mice. Moreover, high levels of phosphorylated form of GAA (PGAA) and reduced amounts of Cr phosphate (PCR) in the heart, skeletal muscle, and brain were detected using ^31^P MRS. Since 2004, the GAMT-D mouse model has been the object of further studies. This review will focus on the main biochemical features in different tissues, especially the brain.

### 3.1. Biochemical Profiling

#### 3.1.1. Bodily Fluids

Guanidine compound concentrations have been evaluated in the plasma/serum and urine of GAMT KO mice and compared with wild-type (WT) and heterozygous (HZ) mice [5,39]. In body fluids, much higher GAA levels in GAMT mice compared to controls were found, while Cr and Crn values were higher in control mice compared to GAMT-D mice (Table 1). The range of Cr and Crn decrease and GAA increase in GAMT-D mice is comparable with human patients [8,11,16]. Table 1 also shows more recent data obtained by Iqbal [40]. To obtain a better comparison, urine Cr, Crn, and GAA content have been reverted to nmol/24 h instead of µmol/L, by assuming female C57BL/6J urinary output of 9.9 ± 3.5 mL/100 g BW/day [41] and 20 g animal BW as reported by Iqbal in Figure 1, Supplementary Materials of Ref. [40]. In addition to Cr, Crn, and GAA, additional guanidine compounds and metabolites were evaluated, though no significant differences were found between the analyzed groups [5,40].

No significant differences in Cr, Crn, or GAA concentrations were found between WT and HZ mice. Significant discrepancies may be observed in the values reported by Iqbal et al.—both in GAMT KO and WT mice—in comparison with those reported by Schmidt et al. and Torremans et al., except for plasma GAA and Crn values in KO mice. To date, it is not possible to justify this discrepancy and, therefore, further studies are necessary in order to evaluate the experimental conditions underlying these differences. In particular, it is necessary to evaluate: (1) breeding conditions; (2) diet; (3) when samples were taken; (4) analysis techniques; and (5) sex and age of the animals.

Moreover, in a recent study [42], GAA plasma levels of approximately 1000 µM were detected in GAMT KO mice—a value at least 10 times higher than that reported by the authors cited in Table 1, further supporting the need to confirm data with different analysis methods. 

#### 3.1.2. Skeletal Muscle

Biochemical hallmarks of GAMT KO muscle tissue are generally characterized by a significant increase in GAA concentration and reduced Cr levels with respect to controls [5]. In addition, the presence of the PGAA and reduced PCR/ATP ratios have been observed [38,43].

MRS is a non-invasive bio-imaging tool used to investigate metabolites in vivo. MRS is largely exploited to analyze both muscle and brain metabolites in GAMT KO mice. In particular, the ^1^H MR-sensitive nucleus has been used to evaluate PCR and choline compounds, such as N-acetyl aspartate, taurine, lipids, etc. In contrast to human studies, PGAA is hard to detect in vivo with ^1^H MR, since it becomes visible on murine muscle as a broad line only when Cr is almost completely depleted [38]. Instead, the ^31^P MR-sensitive nucleus has been used to detect PCR, PGAA, and ATP, mostly expressed as relative ratios. Before reviewing the results obtained by analyzing the MR spectra of muscle tissue as a whole, it should be emphasized that guanidine compounds were also evaluated in the muscle of GAMT WT, HZ, and KO mice on trichloroacetic acid (TCA)-extracted tissues analyzed using a suitable adapted amino acid analyzer [5]. Highly significant differences in GAA and Cr levels were observed between GAMT KO mice and controls. In particular, 14,450 ± 2088 nmol GAA/g tissue was found in KO mice versus 1.88 ± 0.43 and 4.47 ± 1.0 nmol/g in WT and HZ mice, respectively, reflecting GAA values thousands of times higher than control values. Cr was present in GAMT KO mice at levels about 10 times lower than those observed in WT and HZ mice (1501 ± 301 nmol/g tissue vs. 16,508 ± 278 and 18,788 ± 2157, respectively). In total, 13 guanidine compounds were evaluated by Torremans et al. [5], and other compounds—such as Crn, homoarginine, β-guanidinopropionic acid, and γ-guanidinobutyric acid—were significantly higher in the muscle of GAMT KO mice compared to controls. In Table 2 and Table 3, results from in vivo ^1^H and ^31^P MRS analyses on the skeletal muscle of GAMT KO compared to WT mice are reported [38,43,44]. Table 2 also includes the aforementioned Cr and GAA values [5], reverted to mM by assuming a skeletal muscle tissue density of 1.06 g/cc [45].

In addition, in Table 3, PCR and PGAA are expressed both as PCR/ATP and PGAA/ATP, and reverted to mM, starting from ATP concentrations chemically determined by Kan et al. (11.1 ± 1.2 µmol/g dry weight and 15.6 ± 1.1 µmol/g dry weight in GAMT KO and WT mice, respectively) [43], and by assuming the ratio of dry weight/wet weight to be 0.3:0.7 [46].

Residual Cr in GAMT KO muscle is likely caused by diet (creatine-free or not) or coprophagy if animals are not separated as soon as possible from dams and housed in individual cages, depending on the genotype [39,43]. However, Cr is always significantly lower than that observed in control mice, suggesting high-energy phosphate depletion in skeletal muscle. Of interest, the high PGAA/PCR ratio detected (3.4 ± 3) by Renema et al. [38] is probably due to: (1) active GAA uptake via the highly-expressed creatine transporter in muscle tissue, which is also able to transport GAA [47]; (2) endogenous GAA synthesis by *AGAT* [48]; and (3) the ability of creatine kinase (CK) to also use GAA as a substrate [49]. Despite the lower affinity of CK for PGAA, the latter compound functions surprisingly well in vivo as an energy buffer, as shown in a mild ischemic stress experiment [43]. In addition, PGAA accumulation in GAMT KO mice has recently been suggested as a probable explanation for less statin-induced muscle damage in GAMT-D mice compared to *AGAT*-deficient mice [50].

Moreover, marked in vitro [43] and in vivo [39] ATP changes were not present, leading to the hypothesis of adaptive changes in oxidative phosphorylation and in the mitochondria. In the mitochondria of GAMT KO compared to WT mice, Schmidt et al. [39] found a significant increase (of 67%) in the absolute activity of ATP synthase—complex V of the respiratory chain and a marker of aerobic capacity (confirming previous observations by Ilas et al., 2000 [51] in human fibroblasts of GAMT-D patients)—higher (although not significantly) absolute activity of citrate synthase—a marker of mitochondrial content. However, despite the lack of PCR, the accumulation of PGAA, and some adaptive changes, no morphological alterations were observed via electron microscopy analysis of muscle tissue. More recently, metabolic marker enzymes have been investigated in the hind leg muscles of both *AGAT* and GAMT KO mice [52]. In particular, citrate synthase and cytochrome oxidase—complex IV of the respiratory chain—have been evaluated in the gastrocnemius (GAS), plantaris (PLA), and soleus (SOL) muscles, representing glycolytic, intermediate, and oxidative muscles, respectively. In GAMT KO mice, citrate synthase activity and cytochrome oxidase activity were higher in all three muscles compared to controls, confirming previous findings by Schmidt [39]. Moreover, both enzymes had significantly higher activity in males compared to females (citrate synthase in PLA and SOL, cytochrome oxidase in GAS and SOL), suggesting the importance of considering differences between muscles, as well as between males and females, when characterizing phenotypes. However, the changes in GAMT KO mice were modest compared to those in *AGAT* KO mice. The less severe phenotype of GAMT KO mice, as compared to human GAMT-D, could relate to the fact that they accumulate GAA, which functions as a substrate for cytosolic CK, albeit at a reaction rate 100 times slower [49]. According to the authors, CK involvement in the reaction with GAA may prevent activation of AMP-activated kinase (AMP-K) and, in turn, the stimulation of mitochondrial biogenesis, as in *AGAT* KO mice. In addition, since AMP-K is an inhibitor of anabolic pathway activity, its higher activity in this last animal model could also explain the observed muscle atrophy, which is only modest in GAMT KO mice.

Altogether, biochemical features of the skeletal muscle observed in GAMT KO mice—especially those observed with in vivo 31P MRS and 1H MRS—are similar to those observed in human GAMT-D [53].

#### 3.1.3. Brain

The discovery of primary CDS caused by mutations in the genes encoding the *AGAT* [2] and GAMT enzymes [3], or the creatine transporter *SLC6A8* [4], has shed light on the role of creatine synthesis, metabolism, and transport in psychomotor development and cognitive function—in particular in the CNS, which appears to be the main tissue affected by these creatine deficiencies. 

Concerning the GAMT KO brain, most of the authors who have already been cited for their studies carried out on the skeletal muscle of GAMT KO mice have also evaluated the metabolic features of the brain. Renema et al. [38] used 1H MRS to show that the Cr signal (Cr + PCR) was significantly reduced in the brains of GAMT KO mice. Interestingly, Cr reduction was more marked in the brain (83%) than in the muscles (69%), suggesting a slower uptake of orally-ingested Cr in the brain. Indeed, it was observed that only a small amount of Cr reaches the CNS through the blood–brain barrier (BBB) in rats [54], and thus the CNS must rely on endogenous synthesis of Cr for its needs. Similar values have been reported by other authors (Table 4). Recently, Sinha et al. [55] evaluated Cr and GAA content in the brains of GAMT KO mice, confirming the expected biochemical phenotype.

In the ^31^P MR spectra of GAMT KO mouse vs. control mouse brains, a strongly reduced PCR/ATP signal ratio was observed by Renema et al. [38]. In addition, a PGAA/ATP signal ratio was detected, although at a much lower intensity than in muscles (Table 5). Table 5 shows PCR and PGAA concentrations (mM) in GAMT KO mice as estimated with respect to an assumed ATP level of 3 mM [57], as suggested by Renema et al. [38]. To our knowledge, the total ATP content of the GAMT KO mouse brains is unknown.

A similar trend has been observed by Schmidt et al. [39]. Altogether, these results reveal a striking similarity to the biochemical findings in the brains of human patients [3,10,24,58,59,60,61,62].

Significant adaptive changes in ATP synthase activity have also been observed in the mitochondria of the brains of GAMT KO mice, with an increase of 80%. When normalized to citrate synthase, enzyme activity was still significantly higher in the brain, in contrast to findings in muscle tissue [39]. Moreover, Torremans et al. [5] found significant differences in the concentrations of other guanidine compounds aside from Cr, Crn, and GAA, including higher levels of guanidinosuccinic acid (GSA), argininic acid (ArgA), β-guanidinopropionic acid (β-GPA), γ-guanidinobutyric acid (γ-GBA), and homoarginine (Harg) in GAMT KO mice compared to controls. While some differences were similarly observed in the CSF of human patients (γ-GBA and Harg, in addition to the well-known Cr, Crn, and GAA), the increase in brain GSA levels in GAMT KO mice was more pronounced than in human CSF, and the low arginine levels seen in human patients [11] were not observed in mice. These discrepancies found in the brains of GAMT KO mice, as compared with human CSF, could explain the lack of severe neurological symptoms such as epilepsy and ataxia observed in these mice [39]. Indeed, although GAA might be involved in epilepsy in GAMT-D patients [63,64], and although it was present at high levels in the brains of KO mice, epileptic seizures were not observed, suggesting a possible role of other guanidine compounds in helping the brain adapt to the biochemical phenotype due to GAMT-D. In addition, PGAA—a high-energy phosphate able to replace PCR—has long been proposed to explain the mild neurological symptoms observed in GAMT KO mice. Indeed, an increase in PGAA has also been reported in GAMT-D patients [8,65], and a normal amount of this metabolite in the brain is restored by creatine supplementation [65].

In addition, a recent study of partial GAMT-D by RNAi in a 3D model of organotypic reaggregated rat brain cell cultures [66] demonstrated that mild GAA accumulation alone had important consequences on neuron development. Indeed, the authors observed: (1) axonal hypersprouting at early stages that continued up to mature developmental stages; (2) inhibition of natural apoptosis, followed by the induction of non-apoptotic cell death at later stages; and (3) increased expression levels of both GABA_A_ receptors and glutamic acid decarboxylase. These features must be taken into account in future studies aimed at deepening knowledge of the pathophysiology of the brains of GAMT KO mice.

More recently [55], studies on brain morphology using high-resolution MRI have been performed in GAMT KO mice. Four brain structures—the hypothalamus, corpus callosum, internal capsule, and fimbria—were found to be abnormal, both in GAMT KO and *AGAT* KO mice, leading to the conclusion that Cr depletion was the cause of these anatomical changes.

### 3.2. Behavioral Profiling

To assess the quality of an animal model, authors usually refer to a set of validity criteria. In one of the most cited papers regarding animal model validity, Willner described three criteria that should be fulfilled by a good animal model: (1) construct validity (whether the model and human condition have the same trigger causes); (2) face validity (whether the model shows clinical features typical of the human condition); and (3) predictive validity (whether the model responds to treatment in a similar way to humans) [67]. Analysis of the literature shows good construct validity of the GAMT KO model (KO mice are obtained through the inactivation of the murine *GAMT* gene), whereas face and predictive validity are still uncertain. In fact, although typical biochemical human GAMT-D features have been reported in the serum, urine, brains, skeletal muscle, and hearts of GAMT KO mice, only limited behavioral alterations have been demonstrated, and this is probably due to a limited number of studies addressing GAMT KO behavior phenotyping [5,39,40,68,69]. Studies on the behavior assay of the GAMT-D genetic mouse model are summarized in Figure 1.

#### 3.2.1. Early Postnatal Behavioral Development

Although GAMT-D, like phenylketonuria, is a metabolic neurodevelopmental disorder, to the best of our knowledge no data are available on the behavioral phenotype of GAMT-D mice during early postnatal life, since behavioral studies have focused on adult animals [5,39,40,42,68,69]. Schmidt et al. (2004) [39] demonstrated consistent weight differences in female and male GAMT KO mice in comparison with control littermates from the first weeks of life. Moreover, although KO mice are smaller (due to reduced body fat content and muscular hypotonia), no significant difference in length has been reported in pups or adult mice. In addition to body weight and length analysis, studies investigating potential early-onset behavioral deficits in GAMT KO mice are not available. A large set of behavioral assays in both males and females of healthy and mutant mice should be performed in order to identify developmental delay. Communicative behavior, developmental milestone acquisition, neonatal social, maternal, and object recognition, and motor skill development are just some of the typical analyses performed on animal models of neurodevelopmental disorders (such as [70]).

#### 3.2.2. Neuromuscular Disorder

Correct Cr metabolism is important for psychomotor development, and reduced Cr in GAMT-D causes biochemical alterations in the skeletal muscle. For this reason, several studies have explored movement disorders in GAMT KO mice. Analysis of home cage activity [5,39], results from the open field test [5,40], and voluntary exercise capacity tested by providing running wheels in the home cage [69] did not reveal significant or relevant differences in spontaneous motor activity between healthy and KO mice, with the exception of the observation that GAMT KO mice entered corners less than heterozygous mice in the open field test, though the authors acknowledge that this observation is difficult to interpret [5]. No effect of genotype was observed in the accelerating rotarod test—an apparatus that tests equilibrium and motor coordination in mice [5,40,42]. Surprisingly, during forced continuous and accelerated running on a treadmill, in which animals learn to run to avoid electric foot shocks, GAMT KO mice showed better performance than healthy mice, running further and gaining more vertical distance [69]. Conversely, analysis of mean grip force using a grip strength meter test revealed impaired neuromotor performance in GAMT KO vs. control mice [5,39,42].

#### 3.2.3. Learning and Memory Tasks

As previously reported, intellectual disability is the clinical hallmark of GAMT-D, being present in all affected patients. One of the most relevant studies of cognitive performance in GAMT KO mice was performed by Torremans et al. (2005) [5]. Control, KO, and HZ GAMT-D male mice were subjected to a battery of behavioral tests, starting from the age of six months. In addition to motor analysis in the home cage, WT, KO, and HZ mice were subjected to the Morris water maze [5,40]—a test widely used to study visual spatial memory and learning in rodents (mice and especially rats). Animals are placed in a circular pool of opaque water where they have to swim to reach a hidden escape platform using external maze cues. After training, animals reach the platform faster. Torremans et al. [5] reported no performance differences between groups during the training phase of the Morris water maze test, nor in swimming velocity during the test trial. However, during the test trial GAMT KO mice spent significantly less time in the target quadrant containing the platform compared to healthy and HZ mice (although the frequency of entries to this quadrant did not differ between groups). Torremans et al. also explored other forms of learning and memory using the passive avoidance task—a fear-based test used to evaluate the ability of mice to avoid the preferred dark compartment in which a slight foot shock stimulus was previously delivered. Again, no significant differences were found between groups.

In genetic mouse models of metabolic disease, it is necessary to use cognitive tests that do not require food restriction/reinforcement or lengthy training. For example, the spatial novelty test (designed to estimate the ability of rodents to encode spatial relationships [71,72]) and the object recognition test (a variant of the delayed non-match-to-sample task that may be solved in the absence of spatial information [73,74,75]) are recommended because they take advantage of the spontaneous preference that rodents display for novel situations, and are thus appropriate behavioral tests.

In conclusion, due to the limited number of studies on cognitive performance in GAMT-D genetic mouse models, no definitive conclusions can be drawn.

#### 3.2.4. Social Behavior Tasks

Several autistic-like features are frequently reported in the GAMT-D human condition, such as the absence of active speech, hyperactivity, aggressive behavior, self-injurious behavior, excessive interest in objects over social interaction, and stereotypic behaviors.

Regarding sociability, to the best of our knowledge only one study [5] reported no significant differences between healthy, KO, and HZ female mice in the social interaction test. Literature on mouse behavioral tasks relevant to autistic symptoms has grown markedly in the last decades (for a review see [76]). 

Social impairments can be detected by the three-chamber social interaction test used to quantify social behavior deficits in transgenic mice exhibiting autistic-like traits. In particular, the test takes advantage of the natural preference of rodents for social interaction and novelty to investigate in two following sessions whether an animal explores an unknown conspecific more than an inanimate object, and whether it shows a preference for interacting with a familiar rodent or an unknown intruder.

With regard to the absence of active speech, communication deficits have not been explored in GAMT KO mice until now. Although mice do not use language, they emit ultrasonic vocalizations to communicate in specific situations (ultrasounds are displayed by pups to elicit maternal care, by juvenile mice during social play, by adult males during courting, etc.) and, thus, quantitative analysis of ultrasonic vocalizations could allow the assessment of deficits in communication skills. 

In relation to stereotyped behaviors, mice spontaneously exhibit motor stereotypes—such as circling, jumping, backflips, and self-grooming—that are investigated by observing videotaped or real-time sessions, typically during free exploration of an area. In studies in which animals were exposed to the open field test or in which home cage activity was monitored [5,39,40], results regarding these behavioral patterns are not reported. In the article by Iqbal, a detailed behavioral analysis in the open field apparatus is reported in Supplementary Table S2, but it refers to female GAMT KO mice after 10 weeks of dietary supplementation. Perseveration in the exploration of only one part of a test (e.g., only one object in the object recognition test or only an arm in the delayed alternation test) instead of the normal strategy of exploring the whole environment may be considered analogous to the restricted interest observed in human subjects with autism.

In conclusion, in our opinion behavioral phenotyping of the GAMT-D genetic mouse model is not yet complete. An accurate analysis of behavior during early postnatal development, and better characterization of cognitive function and evaluation of stereotyped behavior, would help to better describe possible deficits in GAMT-D mice, thus increasing the validity of this animal model.

## 4. Discussion

The discovery of primary disorders of Cr metabolism and transport has helped clarify the crucial role of creatine in brain metabolism and development. GAMT defects present with early-onset encephalopathy with severe developmental delay/intellectual disability, epilepsy, and movement disorders. From a pathogenetic point of view, this condition results in brain creatine depletion coupled with the accumulation of neurotoxic GAA. Among emerging higher cortical functions, language development is particularly affected by creatine defects. Behavioral disorders in GAMT-deficient subjects have not been well characterized from a clinical point of view, and are usually associated with severe intellectual disability. Like other metabolic neurodevelopmental disorders, such as phenylketonuria, early diagnosis and timely treatment can prevent clinical expression of the disease. Although over 95% of the whole creatine pool is in the muscle tissue, human disease is essentially characterized by a severe and self-limiting derangement of CNS development and functioning. In contrast, the mouse model of GAMT disorders, which shows a similar pattern of biochemical alterations in the muscle and brain, presents with a prevalent neuromuscular disease with minor or no cognitive impairment. This discrepancy between human and mouse models in terms of neurological impairment is currently unexplained. The murine brain seems less vulnerable to both GAA accumulation (no epilepsy) and Cr depletion (no severe cognitive impairment), though there is no conclusive evidence that GAA as PGAA may replace the shuttle functions of Cr in mice differently than in humans. Studies that aimed to explore possible adaptive metabolic variations in GAMT KO mice were inconclusive. While subtle cognitive impairment cannot be ruled out on the basis of present data, the severe intellectual impairment observed in children with GAMT defects is certainly not part of the GAMT KO mouse phenotype.

This is not unusual in mouse models of human diseases. For example, two different mouse models of phenylketonuria, which were developed in strains with different genetic backgrounds (BTBR and C57Bl/6), showed clear differences in behavioral outcome despite comparable biochemical changes in phenylalanine and neurotransmitter concentrations [77]. However, in this case, cognitive impairment severity in the most affected mice (BTBR) also did not replicate that observed in untreated human disease.

The more protracted developmental course of many neurological functions, and the related nervous structure maturation, may be another relevant factor influencing the outcome of early exposure to a neurotoxic noxa in mice and humans.

While awaiting a more systematic assessment of cognitive and behavioral functions, mice with GAMT defects remain an excellent model for the development of new potential treatments targeting the normalization of the metabolic alterations associated with this condition. Indeed, lowering blood GAA coupled with creatine supplementation proved to prevent the derangement of neurological development observed in children with untreated or late-treated GAMT defects [78].

## Figures and Tables

**Figure 1 genes-12-01201-f001:**
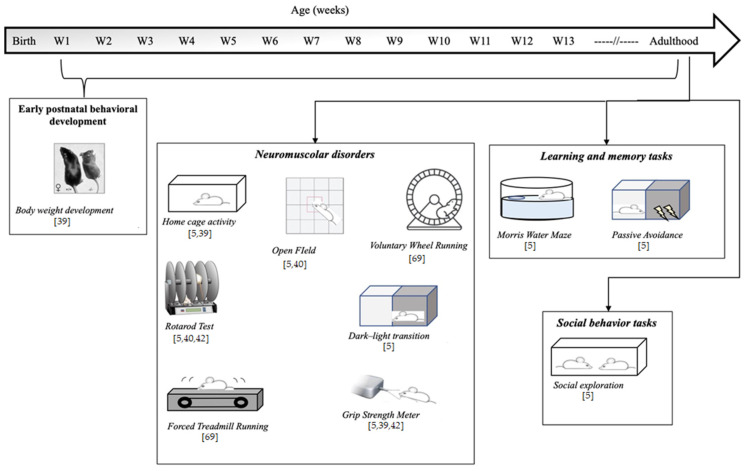
Schematic representation of behavioral tests reported in literature on GAMT knockout (KO), the genetic murine model of GAMT-D.

**Table 1 genes-12-01201-t001:** Cr, Crn, and GAA concentrations in the urine and plasma or serum of GAMT WT, HT, and KO mice.

Tissue	Compound	Genotype							
		WT			HZ		KO		
Urine (nmol/24 h)	Cr	13,490 ± 3967	8394 ± 2739	20,883 ± 7603	5130 ± 1373	5672 ± 1203	71 ± 56 **	47 ± 36 ***	4144 ± 1496
	Crn	9188 ± 1753	6491 ± 1336	54.8 ± 12	6991 ± 853	6206 ± 926	1153 ± 268 ***	536 ± 124 ***	55.6 ± 19.8
	GAA	3005 ± 880	2023 ± 575	320 ± 37	2335 ± 538	2065 ± 522	25,775 ± 2867 ***	19,015 ± 2887 ***	372 ± 27.4
Plasma or Serum (µmol/L)	Cr	172.4 ± 16.4	172 ± 16	1347 ± 205	187 ± 32.7	190 ± 24	12.9 ± 8.1 ***	11.56 ± 7.1 ***	822 ± 416
	Crn	8.9 ± 0.8	8.89 ± 0.73	5.5 ± 0.8	9.4 ± 0.6	9.50 ± 0.94	1.3 ± 0.3 ***	<0.4–2.15 °	1 ± 0.2
	GAA	1.9 ± 0.5	1.94 ± 0.41	33.3 ± 14.6	2.4 ± 0.3	1.97 ± 0.37	117.4 ± 32.1 **	106 ± 29 *	97.6 ± 7.8
Reference		[39] ^A^	[5] ^B^	[40] ^C^	[39] ^A^	[5] ^B^	[39] ^A^	[5] ^B^	[40] ^C^

^A^ Values are expressed as mean ± standard error of the mean (S.E.M). The mice analyzed included 6 WT, 8 HZ, and 10 KO. Statistical significance levels: ** *p* < 0.01, *** *p* < 0.001 (two-tailed heteroscedastic *t*-test). ^B^ Values are expressed as mean ± S.E.M. The mice analyzed included 11 WT, 9 HZ, and 16 KO for urine and 5 WT, 7 HZ, and 11 KO for plasma. Asterisks indicate significant differences compared to the WT genotype (one-way ANOVA and Tukey’s post hoc test: * *p* < 0.05, *** *p* ≤ 0.001; ° indicates that a difference was observed but no statistical test could be performed because of values below the detection limit (DL)). ^C^ Values are expressed as mean ± S.E.M. There were 10 mice for each group (WT and KO).

**Table 2 genes-12-01201-t002:** Cr and GAA concentrations in the skeletal muscle of GAMT-D WT and KO mice.

Tissue	Compound	Genotype					
		WT			KO		
Muscle (mM)	Cr	28.4 ± 2.6	17.5 ± 0.3	27.7 ± 3.9	8.9 ± 3.8	1.6 ± 0.3 ***	3.4 ± 3.9 *
	GAA	-	0.002 ± 0.0005	-	-	15.3 ± 2.2 ***	-
Reference		[38] ^A^	[5] ^B^	[44] ^C^	[38] ^A^	[5] ^B^	[44] ^C^

^A^ There were 5 WT and 7 KO mice. ^B^ Values are expressed as mean ± S.E.M. There were 4 WT and 6 KO mice. Asterisks indicate significant differences compared to the WT genotype (two-way ANOVA and Tukey’s post hoc test: *** *p* ≤ 0.001). ^C^ Values are expressed as mean ± standard deviation. There were 4 WT and 9 KO mice. * Significantly different from WT, with *p* < 0.05. Cr and GAA are referred to as total pools.

**Table 3 genes-12-01201-t003:** PCR and PGAA in the skeletal muscle of GAMT-D WT and KO mice.

Tissue	Compound	Genotype			
		WT		KO	
Muscle (phosphate ratios)	PCR/ATP	2.66 ± 0.11	3.16 ± 0.10	0.76 ± 0.33	N.D.
	PGAA/ATP	N.D.	N.D.	1.78 ± 0.42	3.04 ± 0.06
Muscle (mM)	PCR	18.86 ± 0.78	22.40 ± 0.71	3.83 ± 1.66	N.D.
	PGAA	N.D.	N.D.	8.97 ± 2.01	15.32 ± 0.30
Reference		[38]	[43]	[38]	[43]

All values are presented as mean ± standard deviation. As suggested [46], PCR/NTP and PGAA/NTP ratios in Renema et al. are reported as PCR/ATP and PGAA/ATP ratios. N.D.: not detectable.

**Table 4 genes-12-01201-t004:** Cr, Crn and GAA concentrations in the brain tissue of GAMT-D WT, HZ and KO mice.

Tissue	Compound	Genotype							
		WT			HZ		KO		
Brain (mM)	Cr	10.752 ± 1.15	11.841 ± 0.44	8.2 ± 1.2	11.005 ± 0.439	10.853 ± 0.488	0.454 ± 0.097	0.489 ± 0.09 ***	1.4 ± 0.4
	Crn	0.356 ± 0.04	0.339 ± 0.50		0.186 ± 0.049	0.258 ± 0.040	0.00945 ± 0.00004	<DL-0.02121 °	
	GAA	0.0123 ± 0.002	0.0128 ± 1.37		0.0148 ± 0.001	0.0134 ± 0.001	1.958 ± 0.070	1.945 ± 0.064 ***	1.3 ^
Reference		[39] ^A^	[5] ^B^	[38] ^C^	[39] ^A^	[5] ^B^	[39] ^A^	[5] ^B^	[38] ^C^

^A, B^ Brain density 1.05 g/cm^3^ [56]. ^B^ Asterisks indicate significant differences compared to WT genotype (two-way ANOVA and post hoc Tukey: *** *p* ≤ 0.001; ° indicates that difference is observed but no statistical test could be done because of values below detection limit). ^C^ Creatine + phosphocreatine. ^ Renema et al. calculated values assuming that brain tissue ATP concentration was also ~3 mM in GAMT mice, and that PGAA levels in the brain estimated from signal ratios were around 1 mM. Assuming that about 75% of the total pool is phosphorylated (as in Cr), the total GAA pool in the brain would be of the order indicated above.

**Table 5 genes-12-01201-t005:** PCR and PGAA in the brain tissue of GAMT-D WT and KO mice.

Tissue	Compound	Genotype	
		WT	KO
Brain (phosphate ratios)	PCR/ATP	1.36 ± 0.46	0.31 ± 0.10
	PGAA/ATP	N.D.	0.28 ± 0.07
Brain (mM)	PCR	4.08 ± 1.38	0.93 ± 0.3
	PGAA	N.D.	0.84 ± 0.21
Reference		[38]	

PCR/NTP and PGAA/NTP ratios in Renema et al. are considered as PCR/ATP and PGAA/ATP ratios as suggested [46]. N.D. = not detectable.

## Data Availability

Not applicable.

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
