# Peer review of "Intellectual Disability and Brain Creatine Deficit: Phenotyping of the Genetic Mouse Model for GAMT Deficiency"

_genes, 2021, doi:10.3390/genes12081201_

Round 1

Reviewer 1 Report

This is a complete and rather straightforward review of an animal model of GAMT deficiency. The discussion of the animal model itself is complete and thorough. There are few minor changes associated with its presentation that I believe are necessary and would benefit the paper. The major addition that I believe is needed prior to publication is a more complete description of the human disease phenotype that the mouse KO is attempting to study.

The value in this paper is in summarizing how exactly the mouse model does and does not mimic the human disease. This should be more explicitly drawn out. 

With regards to the human disease, I believe two major discussion points are lacking. The first is a review of the genetic causes of GAMT-D. What mutations are associated with the disease? Are they recessive KOs as well? Heterozygous KOs? Full KOs? Partial KOs? Dominant negatives? For a complete review of the model it is important that the genetic origins of the disease itself are clear.  Similarly, in each section on the biochemical/physiological profiling of the mouse KO model, the authors write something to the effect of "comparable to human patients" (e.g., lines 153-154, 265-266, 300-302). To be most meaningful it would be extremely helpful to present the data for humans as well. This could be as a table in section 2 or as additions to the various tables in section 3. Regardless I think that it is necessary.

Smaller points that I think should be addressed include identifying significant changes in Table 1 associated with Schmidt et al HZ urine creatine and all KO values, and with the Iqbal et al. KO urine creatine values. I also believe that significantly more discussion needs to be included on the Iqbal et al study that produces such wildly different values compared to Schmidt and Torremans. 

Finally, I think that in Figure 1 and associated sections of section 3.2 "Intellectual disability" tests in rodents should be referred to as "learning and memory tasks" and "autistic-like behaviors" be referred to as "social behavior tasks". 

Author Response

First of all, we would like to thanks the Reviewer#1 for his valuable comments.

Point 1: The value in this paper is in summarizing how exactly the mouse model does and does not mimic the human disease. This should be more explicitly drawn out. With regards to the human disease, I believe two major discussion points are lacking. The first is a review of the genetic causes of GAMT-D. What mutations are associated with the disease? Are they recessive KOs as well? Heterozygous KOs? Full KOs? Partial KOs? Dominant negatives? For a complete review of the model it is important that the genetic origins of the disease itself are clear.  

Response 1: Pattern of inheritance of GAMT-D and known pathogenic variants of the disease have been added in the relevant section (2. GAMT-D syndrome: clinical and biochemical profiling).

Point 2: Similarly, in each section on the biochemical/physiological profiling of the mouse KO model, the authors write something to the effect of "comparable to human patients" (e.g., lines 153-154, 265-266, 300-302). To be most meaningful it would be extremely helpful to present the data for humans as well. This could be as a table in section 2 or as additions to the various tables in section 3.

Response 2: Biochemical profile of GAMT-D patients has been added in the relevant section (2. GAMT-D syndrome: clinical and biochemical profiling).

Point 3: Smaller points that I think should be addressed include identifying significant changes in Table 1 associated with Schmidt et al HZ urine creatine and all KO values, and with the Iqbal et al. KO urine creatine values. I also believe that significantly more discussion needs to be included on the Iqbal et al study that produces such wildly different values compared to Schmidt and Torremans.

Response 3: Significant changes in Table 1 associated with Schmidt et al. KO values have been added. Significant changes in Table 1 associated with Schmidt et al. HZ  values have not been added since Schmidt at al. at page 909 line 7 wrote: “No significant differences were found in the concentrations of these substances in the three types of samples between wild-type and heterozygous mice”. Three types of samples were referred to: GAA, Cr and Crn content in serum, urine and brain. Significant changes in Iqbal et al. KO urine creatine values are not reported in the Iqbal’s paper (Table 2, pag 1956). Indeed, P values are referred to KO mice supplemented with 2% creatine or creatine free diet and not to wild-type and KO mice. Iqbal’s data are in agreement with Schmidt and Torremans only concerning creatinine in wild-type and KO mice, and GAA in KO mice. This has now been added in the text. The differences in values found in Iqbal’s study as compared to Schmidt and Torremans studies has not been discussed since, in our opinion, some different experimental conditions used by Iqbal (younger mice and only female mice) don’t support the observed discrepancies. We further think that analysis techniques to evaluate guanidine compounds, in particular mass spectrometry, needs to be further investigated. 

Point 4: Finally, I think that in Figure 1 and associated sections of section 3.2 "Intellectual disability" tests in rodents should be referred to as "learning and memory tasks" and "autistic-like behaviors" be referred to as "social behavior tasks".

Response 4: The terms have been replaced as required, in the text and in the figure.

Reviewer 2 Report

Hi Authors,

Thank you for your nice and sound work. This review gives us a clear understanding of the Guanidinoacetate methyltransferase deficiency (GAMT-D) genetic mouse model research. I have some minor comments on your paper.

  1. The tables are a little difficult to follow. Make that simple.
  2. For the behavior studies make some clear notes. It seems the most of the cited article doesn’t give the reader about a clear phenotype of the model.
  3. You can put some clinical therapeutics approach regarding the deficiency in conclusion.

Author Response

First of all, we would like to thanks the Reviewer#2 for his kind comments

Point 1: The tables are a little difficult to follow. Make that simple.

Response 1: In order to better follow the tables, lines have been added allowing easier identification of the animals’ groups according to the genotype.

Point 2: For the behavior studies make some clear notes. It seems the most of the cited article doesn’t give the reader about a clear phenotype of the model.

Response 2: as reported in the ms, in our opinion behavioral phenotyping of the GAMT-D genetic mouse model is too partial to depict a phenotype of the model.

Point 3: You can put some clinical therapeutics approach regarding the deficiency in conclusion.

Response 3: Some clinical therapeutics approaches have been added at the end of the section “2. GAMT-D syndrome: clinical and biochemical profiling”. At the end of the discussion section the actual approach is mentioned.